# VRouter: Micro-Batch Level Load Balance via Inter-EP Routing for MoE Training

**#PaperID: 11490**

## Abstract

Load imbalance within the Expert Parallel (EP) group leads to poor GPU efficiency when pre-training large-scale Mixture-of-Experts (MoE) models. Though recent approaches have attempted to mitigate this through dynamic expert rearrangement at the global-batch level, they overlook the rapid and dynamic variations in load distribution across different micro-batches. Additionally, relocating or shadowing popular experts at micro-batch level incurs substantial communication overhead due to frequent migrations of expert parameters and gradients.

To address these issues, we introduce VRouter, a novel Inter-EP routing system that achieves better load balance at the micro-batch level, without requiring any expert migration or replication. We have three key techniques: (1) VRouter utilizes the *expert shifting strategy* that allows workloads to be redistributed across neighboring devices, creating additional opportunities for balancing, (2) VRouter adopts *expert dropping mechanism* to reduce both per-device memory footprint and gradient synchronization overhead across EP groups, by selectively dropping experts while preserving load balance, and (3) VRouter applies a lightweight *load-aware token routing algorithm* that redistributes load across devices uniformly. Experimental evaluations on representative MoE models demonstrate that VRouter achieves 1.05-1.13× throughput speedup over existing routing systems.

## 1 Introduction

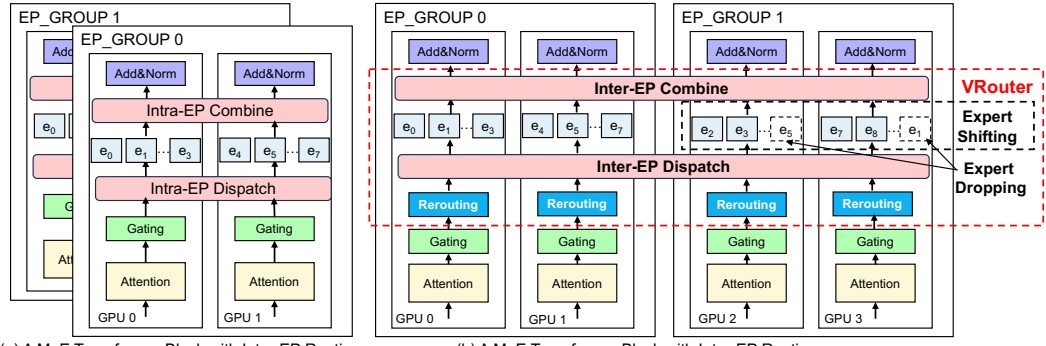

(a) A MoE Transformer Block with Intra-EP Routing      (b) A MoE Transformer Block with Inter-EP Routing

Figure 1: Workflow of a Transformer Block without and with VRouter.

In recent years, large-scale Mixture-of-Experts (MoE) (Liu et al., 2024; Yang et al., 2025; Team et al., 2025) models have shown remarkable performance in natural language reasoning (DeepSeek-AI et al., 2025; Bai et al., 2025) and generation (Vaswani et al., 2017; Fedus et al., 2022; OpenAI, 2023), due to the increasing model capacity. This architectural paradigm replaces conventional feed-forward network (FFN) modules with MoE modules comprising multiple specialized subnetworks termed *experts*, where each input token is dynamically routed to a sparse subset (e.g., Top-$k$ (Zhang et al., 2025)) of experts for computation. Training of large-language models typically requires trillions of tokens, which are first grouped into multiple global batches for iterative computation, and each is further divided into several micro-batches for gradient accumulation (Shoeybi et al., 2019).

Training MoE models on massive datasets requires distributed parallelism strategies. As shown in Figure 1(a), under data parallelism (DP) (Sergeev & Balso, 2018; Li et al., 2020), tokens within a global-batch are evenly partitioned across devices in a DP group, where gradients are locally

accumulated and synchronized only after completing a global-batch. In expert parallelism (EP) (Lepikhin et al., 2020; He et al., 2021), the experts of each layer are sharded across devices in an EP group. During execution, tokens are dynamically dispatched via All-to-All (He et al., 2021; Liu et al., 2024) to the devices hosting their activated experts (termed Intra-EP Routing).

**Micro-batch Level Load Imbalance.** As MoE models become more fine-grained, the characteristics of the workload have largely changed. Qwen team (Qiu et al., 2025b) proposes to ensure load balance across global-batches, while tolerate imbalance across micro-batches for better model quality. However, the micro-batch level imbalance introduces significant system challenges, and it is overlooked by existing works. Specifically, within a single micro-batch, the inherent non-uniform routing distribution causes experts deployed on different GPUs to receive varying numbers of tokens, leading to discrepancies in both computation and communication costs across devices. Therefore, the execution time of the MoE module is dominated by the expert receiving the largest number of tokens. In a real-world workload, we observed that the time cost between the heaviest and lightest loaded devices within the same EP group differed by up to 31.79%, such an imbalance prevents full utilization of GPU resources. Since the duration of micro-batch is short, as well as the imbalance states change frequently and irregularly, it is challenging to predict and exploit these states for effective load-balance strategies.

Existing approaches to load balancing can be broadly categorized into two paradigms, both of which fall short at the micro-batch level. The first class leverages historical routing statistics across global-batches to identify frequently accessed ("hot") experts and applies proactive measures such as expert replication (Qing et al., 2025) or exchanging (Zhai et al., 2023). However, these coarse-grained, global-batch level optimizations fail to capture fine-grained load variations that occur dynamically within micro-batches. The second class reacts to real-time routing outputs and dynamically replicates hot experts during training. For example, FasterMoE (He et al., 2022) broadcasts popular experts to all devices, while FlexMoE (Nie et al., 2023) and SwiftMoE (Skiadopoulos et al., 2025) selectively replicate them. Although more responsive, these methods incur substantial communication overhead due to frequent parameter and gradient transfers—overhead that becomes prohibitive when applied at the micro-batch granularity, where computational duration are inherently narrow.

To address these challenges, we propose VRouter, a novel Inter-EP routing system that achieves better load balance at the micro-batch level, while without requiring any expert migration or replication. Our key insight is that load imbalance can be mitigated not by moving experts, but by enabling *Inter-EP Routing*: allowing tokens to be dispatched across multiple EP groups as shown in Figure 1(b). Building on this mechanism, VRouter further incorporates three key techniques. First, we observe that a significant imbalance arises across different EP groups. Based on this, we propose the *expert shifting strategy*. At the beginning of training, VRouter pairs every two EP groups and cyclically left shifts the expert placement within one group. This shift allows workloads to be redistributed across neighboring devices, creating additional opportunities for balancing. Second, under the Inter-EP Routing, each EP group no longer needs to maintain a complete set of experts to perform the computation. We propose the *expert dropping mechanism* that reduces both per-device memory footprint and gradient synchronization overhead across EP groups by selectively dropping experts while preserving load balance. Finally, VRouter applies a *load-aware token rerouting algorithm* to rewrite the routing map. At the granularity of EP group pairs, it gathers device load distributions and uses greedy search to generate a new map, yielding a more balanced workload.

We evaluate VRouter on representative MoE training workloads and compare it against state-of-the-art baselines. The results show that VRouter achieves up to $1.13\times$ speedup in the end-to-end training throughput, significantly outperforming existing load-balancing techniques.

## 2 BACKGROUND

### 2.1 MoE MODELS AND PARALLELISM STRATEGIES

Mixture-of-Experts (MoE) model is a neural network architecture where multiple feed-forward networks (FFNs, termed experts) are trained in parallel and a learnable router dynamically selects a small subset of experts to process input token. This sparse activation mechanism enables MoE models to scale to trillions of parameters while keeping the per-token computation cost manageable (Bai et al., 2025; DeepSeek-AI et al., 2025).

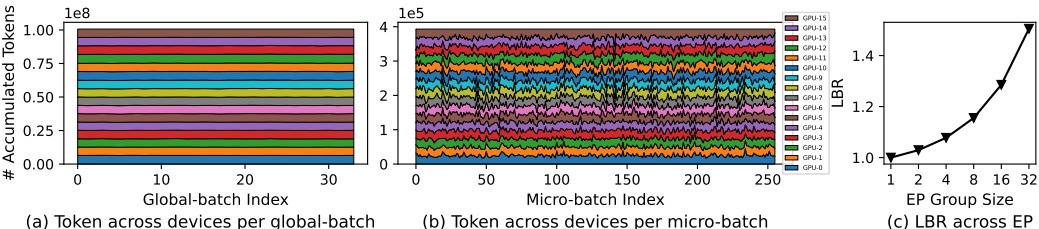

Figure 2: Token distribution at the global-batch level and micro-batch level. We profiled load distribution over 35 consecutive global batches from an EP group, starting at iteration 125,000 on Qwen-S with DP4EP16. Detailed model configurations are listed in Table 2. Colors indicate the token count per device.

To efficiently train MoE models, adopting distributed parallelism strategies is essential. Two widely used strategies are as follows.

**Data Parallelism (DP).** In DP (Li et al., 2020; Paszke et al., 2019), the full model is replicated across multiple devices, and each replica processes a distinct subset of training samples in parallel. After completing a global-batch, the gradients are aggregated before updating the model parameters.

**Expert Parallelism (EP) and Intra-EP Routing.** Recent scaling law analysis (Bai et al., 2025) reveals that increasing sparsity leads to significant performance gains, motivating the design of MoE models with a growing number of experts. To accommodate the rapidly increasing memory demands of these experts, EP (Lepikhin et al., 2020; He et al., 2021) distributes the experts across a dedicated group of devices, referred to as an EP group. Within an EP group, tokens are dispatched via All-to-All communication to devices hosting the target experts. We term this as Intra-EP Routing since the each token can only be assigned to devices inside one EP group. The outputs of experts are combined and passed to subsequent layers.

## 2.2 RELATED WORKS

**Algorithm Optimization.** Load-Balancing Loss (LBL) (Lepikhin et al., 2020; Fedus et al., 2022; Qiu et al., 2025b) is a widely adopted technique in training MoE to encourage balanced expert utilization, which is motivated by the fact that the gating network tends to allocate tokens to a small subset of experts without intervention in expert routing, leading to severe imbalance in expert utilization and degrading the overall performance of MoE models. LBL incorporates a regularization objective that penalizes skewed routing behavior, specifically when an expert receives a disproportionately high number of tokens, thus fostering balanced expert utilization throughout the model.

**System Optimization.** To achieve better load balance across devices, existing works fall into two categories. The first leverages historical routing statistics over global batches to identify frequently accessed hot experts, and applies coarse-grained optimizations such as expert replication (e.g., Hecate (Qing et al., 2025)) or expert exchange (e.g., SmartMoE (Zhai et al., 2023)). The second category adapts real-time routing outputs to dynamically replicate hot experts. For instance, Faster-MoE (He et al., 2022) broadcasts popular experts to all devices, while FlexMoE (Nie et al., 2023) selectively shadows them. They both rely on *global-batch level* information to design balancing strategies.

## 3 MOTIVATION

### 3.1 THE LOAD IMBALANCE AT MICRO-BATCH LEVEL

Prior work focuses on load imbalance across devices at the global-batch level. However, our analysis shows a sharper challenge: imbalance emerges remarkably at the *micro-batch level*, where dynamics are highly volatile and unpredictable. This shift is driven by two factors: (1) the adoption of advanced Global-batch Balance Loss (Qiu et al., 2025a) to improve performance and interpretability of models, and (2) the scaling of MoE models to hundreds of experts (e.g., 256 in DeepSeek-V3 (Liu et al., 2024)) with extreme sparsity (e.g., eight active experts per token), which necessitates high EP degrees to fit the model within multi-device memory.

To illustrate this severe imbalance, we analyze token distribution across devices at both global- and micro-batch levels. For each device, the global-batch token count is defined as the sum of all tokens received across micro-batches in one iteration. As illustrated in Figure 2(a), owing to the Global-

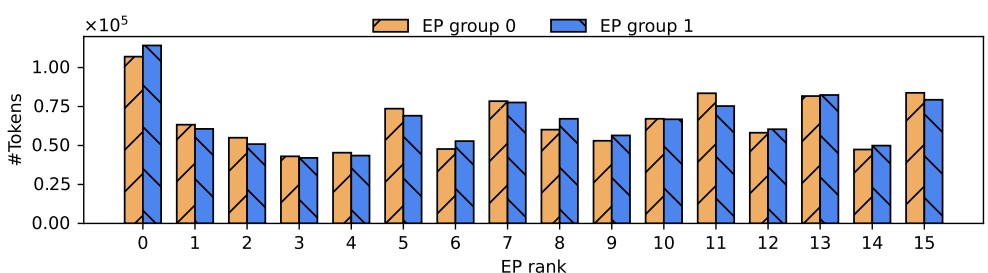

Figure 3: Token distribution in different EP groups, obtained from Qwen-S pretraining with DP2EP16.

batch Balance Loss, token counts remain balanced across devices at the global-batch level. However, load distribution remains highly skewed at the micro-batch level as shown in Figure 2(b), where hot experts fluctuate unpredictably, defying the predictive strategies of prior methods (Qing et al., 2025; Zhai et al., 2023).

To gain a deeper insight into the load imbalance issue, we introduce the Load Balance Ratio ($LBR$), a metric designed to quantify load distribution within each micro batch, defined as,

$$LBR = \frac{\max_{g \in \text{EP\_GROUP}} \left\{ \sum_{(e,g) \in P} n_e \right\}}{\text{AVG\_TOKEN\_PER\_DEVICE}} \tag{1}$$

Formula 1 measures the deviation of actual token reception from the ideal load balance for each layer within an EP group. Given that expert $e$ is assigned with $n_e$ tokens, the token load of device $g$ equals the sum of tokens routed to all experts hosted on $g$. Within an EP group, the device with the largest token load is the straggler, since others must wait for its completion before combination. Therefore, a higher $LBR$ reflects a more severe load imbalance. Figure 2(c) shows the $LBR$ across EP with different EP group size. When the EP group size is 1, the workload is uniformly distributed while the $LBR = 1$. As the EP group size increases, the $LBR$ increase dramatically. When the EP group size reaches 32 (EP32), the $LBR$ is as high as 1.5, which indicates worse balance.

### 3.2 LIMITATIONS OF INTRA-EP ROUTING

Existing Intra-EP Routing approaches, including SmartMoE (Zhai et al., 2023) and FasterMoE (He et al., 2021), fail to address micro-batch imbalance for three key reasons. First, their global-batch level adaptation cannot react to micro-batch volatility. SmartMoE adjusts expert placement based on load skew observed across the entire global batch. This strategy breaks down when the global batch appears balanced overall, yet individual micro-batches exhibit significant imbalance. Second, while FasterMoE dynamically replicates experts based on real-time load, modern MoE models contain so many fine-grained experts that replicating just a few is ineffective—yet replicating many introduces severe memory pressure, risking OOM errors and limiting scalability. Finally, dynamic expert migration, especially across nodes, incurs heavy communication overhead. For example, FasterMoE's replication increases FFN layer time by 78%, mostly from parameter transfer costs, negating any load-balancing gains.

### 4 OPPORTUNITIES AND CHALLENGES

Token routing across EP groups creates new opportunities for fine-grained load balancing. For example, consider eight GPUs with the data-parallel degree of two and expert-parallel degrees of four (Figure 4(a)). GPU0 and GPU4 belong to different EP groups but the same DP replica, so they store identical expert parameters. If GPU0 becomes overloaded, some of its tokens can be routed to GPU4 to relieve the imbalance without moving any expert parameters. After GPU4 finishes its computation, GPU0 fetches and aggregates the returned results, ensuring consistency with the Intra-EP Routing outcome. Compared with rearranging experts every micro-batch as in Intra-EP Routing, Inter-EP Routing provides a flexible way to rebalance token distribution while avoiding frequent, costly expert migrations. However, it is non-trivial to achieve optimal Inter-EP Routing to maximize training efficiency.

**Homogeneity Load Distribution across EP Groups.** A fundamental challenge in realizing the benefits of Inter-EP Routing lies in the inherent similarity of expert load patterns across different

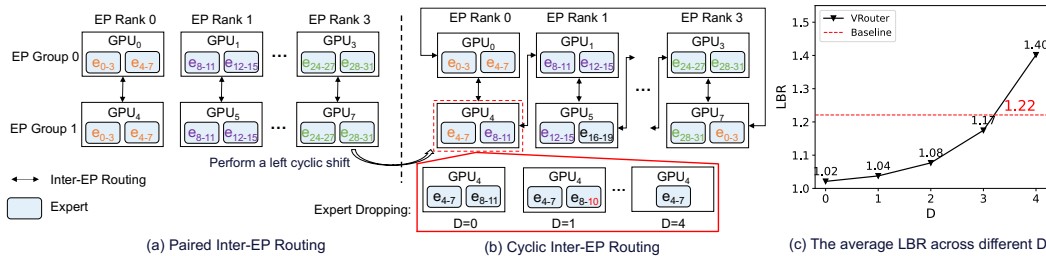

Figure 4: An example of two Inter-EP Routing variants and expert dropping mechanism on $LBR$. The model has 16 experts and employs DP2EP4 as parallelism strategy. (a) Paired Inter-EP Routing before performing a left cyclic experts shift. (b) Cyclic-Inter-EP Routing after the shift and the expert dropping mechanism of VRouter. (c) $LBR$ across different $D$, based on data obtained from Qwen-S pretraining.

EP groups. Since each EP group processes data from the same overall distribution, their gating networks tend to activate similar sets of experts—e.g., the same "hot" experts are simultaneously overloaded across all groups, while others remain underutilized. To validate this, we analyze the expert workload distribution across multiple EP groups in a real MoE training setup with two data parallel (DP) groups in Figure 3. We can observe that devices with the same EP rank across different DP groups exhibit highly similar token load patterns—e.g., both ranks labeled as "EP rank 0" receive significantly more tokens than others, while "EP rank 3" remains relatively underutilized.

As a result, even with the cross-group token dispatch enabled by Inter-EP Routing, there may be limited opportunities for effective load balancing: a device in one EP group cannot offload tokens to a peer in another group if both are experiencing a high load on the same expert dimension. This homogeneity limits the statistical multiplexing gain and poses a key challenge for any Inter-EP Routing system aiming to improve hardware utilization through spatial workload migration.

**Effective and Low Latency Routing Algorithm.** Inter-EP Routing transforms token dispatch from a deterministic lookup into a dynamic, multi-target assignment problem. Unlike Intra-EP Routing, it must balance workload and communication cost across devices while making fine-grained routing decisions at every layer and micro-batch. Crucially, the routing algorithm must be both effective—to avoid creating new imbalances—and low-latency, as it is executed frequently during training. Designing such an algorithm poses a significant challenge in achieving efficiency without sacrificing effectiveness.

## 5 DESIGN OF VROUTER

### 5.1 CYCLIC EXPERT SHIFTING AND EXPERT DROPPING

To fully realize the potential of Inter-EP Routing , we propose a static yet effective expert placement strategy—applied at initialization and fixed throughout training—that enhances routing flexibility without incurring runtime overhead. Our approach combines two key techniques: *Cyclic Expert Shifting* and *Expert Dropping*, which jointly improve load balancing opportunities while reducing memory footprint.

**Cyclic Expert Shifting.** The MoE pretraining system typically organizes different EP groups into a homogeneous pattern. Specifically, it first evenly distributes experts in MoE layers across devices within an EP group, then replicates this expert arrangement to other EP groups via data parallelism. This results in all devices with the same EP rank containing identical expert parameters. As shown in the left half of Figure 4 (a), EP group 0 and EP group 1 adopt identical expert placement patterns. For instance, GPU 0 and 4 (both are EP rank 0) store experts from $expert_0$ to $expert_7$.

Starting from homogeneous expert placement, we cyclically shift experts in EP group 1 by 4 positions (half the number of experts per device), leading to heterogeneous EP groups with interleaved expert placement. This design causes the workloads of the two EP groups to be staggered across different EP ranks. Based on the heterogeneous EP groups, any device is able to offload part of its workload to two other distinct devices, forming a complete ring structure between the two EP groups at the end. This ring-based Inter-EP Routing is termed *Cyclic Inter-EP Dispatch*. As illustrated in Figure 4 (b), EP rank 0 within EP group 1, has two offloading options: it can redirect the workload destined for $expert_{4-7}$ to another EP rank 0, or to $expert_{8-11}$ on EP rank 1.

Table 1: List of Symbols and Definitions

| Symbol | Definition |
|---|---|
| $e$ | An individual expert in the Mixture-of-Experts (MoE) layer. |
| $E$ | The set of all experts within a cyclic EP group, i.e., $E = \{e_1, e_2, \dots\}$. |
| $g$ | A single GPU device. |
| $G$ | The set of GPUs that constitute a cyclic EP group, i.e., $G = \{g_1, g_2, \dots\}$. |
| $P$ | $(e, g) \in P$ indicates that the expert $e$ lies in GPU $g$. |
| $\mathbb{S}[\cdot]$ | Expert mapping function. $\mathbb{S}[e]$ returns the expert in another EP group that shares identical parameters with expert $e$. |
| $\mathbb{P}[\cdot]$ | Expert placement function. $\mathbb{P}[e]$ returns the GPU $g$ on which expert $e$ is located. |
| $I$ | Routing matrix generated by the top-$k$ router. An entry $I_{i,j} = e_0$ indicates that the $j$-th expert selected for the $i$-th token is $e_0$. |

Based on our observations in Section 3, at the granularity of a global-batch, each expert receives a nearly identical number of tokens. Consequently, over the course of training, the expected workload per expert exhibits no significant variation. This implies that, during expert shifting, we do not need to decide which specific expert should be moved to which location. In the experimental section, we demonstrate that such an arrangement effectively alleviates the problem of load imbalance.

**Expert Dropping.** In the Intra-EP Routing scenario, each EP group must host a complete set of experts so that all incoming tokens can be processed locally. In contrast, when Inter-EP Routing is enabled, tokens may be rerouted to another EP group to locate and invoke the target expert even if the originating EP group does not host it, thereby reducing the number of experts that need to reside on each device. Based on this observation, we propose the *Expert Dropping* technique, where users can specify a hyperparameter $D$. The system will drop the last $D$ experts in the Cyclic EP group configuration when initializing EP groups. As shown in Figure 4(c), although $D = 0$ yields the lowest $LBR$ (i.e., the most balanced load), setting $D = 1$ and dropping one expert per device only slightly increases $LBR$ from 1.02 to 1.04, so the load remains well balanced. Dropping experts reduces per-device memory footprint and lowers the synchronization overhead of expert parameters and gradients across DP groups, which can also improve performance. We confirm this in Section 6: after removing some experts, VRouter achieves larger performance gains while saving GPU memory. We can drop useless experts to save GPU memory while improving computational throughput through Inter-EP Routing. It is noteworthy that when half of the experts are removed, Inter-EP Routing degenerates into Intra-EP Routing, and the EP group size becomes twice that of the case when $D = 4$ in the example of Figure 4.

## 5.2 INTER-EP ROUTING ALGORITHM

In this section, we present a lightweight load-aware token rerouting algorithm to construct a new routing map that redistributes load across devices uniformly.

VRouter expands the routing possibilities by allowing tokens to be dispatched to multiple target devices through Inter-EP Routing. While this flexibility expands routing possibilities, it also presents a new challenge: VRouter needs to strategically allocate tokens across devices to maximize system performance. The key goal is to maintain balanced GPU utilization by preventing computational hotspots that arise from uneven expert activations. This requires the routing algorithm to simultaneously consider both the real-time load distribution and expert placement across devices. Since this load balancing occurs at each transformer layer in each micro-batch, the process must be computationally efficient. Therefore, we design a lightweight algorithm that can make rapid routing decisions with negligible overhead. VRouter routes tokens to targeted devices via All-to-All primitive according to the plan.

Algorithm 1 lists the Inter-EP Routing algorithm and Table 1 describes the meaning of symbols used. It takes as input the current local routing plan $I^{in}$ and the expert placement $P$ that specifies where each expert is placed, and produces the optimized routing plan $I^{out}$. Each device first counts how many tokens are assigned to each of the experts for its local input routing plan, and then exchanges these counts across devices to obtain the global view of token distribution (L2-3). Devices are then processed in descending order of their total token load, prioritizing those most in need of offloading.

---

**Algorithm 1** Routing Optimization for Load Balancing

---

**Require:** Input routing map $I^{\text{in}}$, Expert-to-GPU assignment partition $P$
**Ensure:** Optimized routing map $I^{\text{out}}$

1: $I^{\text{out}} \leftarrow I^{\text{in}}$
2: $\mathcal{N}^{\text{local}} \leftarrow \text{CountTokens}(I^{\text{in}})$      ▷ Number of tokens assigned to each expert on this device
3: $\mathcal{N}^{\text{global}} \leftarrow \{\mathcal{N}_g^{\text{local}} | g \in \mathcal{G}\}$      ▷ Collect token counts across all devices in the group
4: **for** each $g \in \text{SortDesc}(\mathcal{G}, \mathcal{N}^{\text{global}})$ **do**      ▷ Process devices in descending order of total load
5:      $\mathcal{G}' \leftarrow \{\mathbb{P}[e'] \mid (e, g) \in P, e' = \mathbb{S}[e]\}$
6:      **for** each $(e, g) \in P$ such that $\mathcal{N}_{g,e}^{\text{global}} > 0$ **do**
7:          $e' \leftarrow \mathbb{S}[e]$
8:          $g' \leftarrow \arg\min_{d \in \mathcal{G}'} \mathcal{N}_d^{\text{global}}$      ▷ Select least-loaded peer device with $e'$
9:          $\Delta \leftarrow \left\lfloor \frac{\mathcal{N}_g^{\text{global}} - \mathcal{N}_{g'}^{\text{global}}}{2} \right\rfloor$      ▷ Max tokens we can balance from $g$ to $g'$
10:         $\Delta' \leftarrow \min\left(\mathcal{N}_{g,e}^{\text{global}}, \Delta\right)$      ▷ Actual number of tokens to reroute
11:         **if** $\Delta' > 0$ **then**
12:             $\mathcal{N}_{g,e}^{\text{global}} \leftarrow \mathcal{N}_{g,e}^{\text{global}} - \Delta'$
13:             $\mathcal{N}_{g',e'}^{\text{global}} \leftarrow \mathcal{N}_{g',e'}^{\text{global}} + \Delta'$
14:             **if** $I^{in}$ locates in $g$ **then**
15:                 $I^{\text{out}} \leftarrow \text{Reroute}(I^{\text{out}}, e \rightarrow e', \Delta')$      ▷ Redirect $\Delta'$ tokens from $e$ to $e'$
16:             **end if**
17:         **end if**
18:      **end for**
19: **end for**
20: **return** $I^{\text{out}}$

---

Table 2: Configurations of the evaluated models

| Model Name | Hidden Size | Intermediate Size | #Layer | #Expert | Top-K |
|---|---|---|---|---|---|
| Qwen-S | 2048 | 768 | 24 | 128 | 6 |
| Qwen-L | 4096 | 1024 | 8 | 128 | 8 |
| Qwen-XL | 8192 | 2048 | 4 | 512 | 8 |
| Qwen-XXL | 8192 | 2048 | 12 | 512 | 8 |

For each expert $e$ on a heavily loaded device $g$, the algorithm finds the corresponding peer expert $e'$ located on another device using the expert mapping function $\mathbb{S}[\cdot]$ (L5). Among all devices that hosts $e'$, it selects the least-loaded one, denoted as $g'$. The number of tokens to transfer from $g$ to $g'$ is set to the minimum between (i) half of the load difference between the two devices and (ii) the number of tokens currently assigned to expert $e$ on $g$ (L7-10). If the computed transfer amount $\Delta' > 0$, the algorithm updates the global token counts for both experts and performs a rerouting operation where $\Delta'$ tokens originally assigned to $e$ are redirected to $e'$ in $I^{out}$ if $g$ is the current device where $I^{in}$ locates in (L11-16). This process repeats for all experts on the selected device and continues down the sorted list of devices, ultimately producing an optimized and balanced routing plan $I^{out}$.

## 6 EXPERIMENTS

### 6.1 EXPERIMENTAL SETUPS

**Testbed.** We evaluate our approach on two NVIDIA GPU (NVIDIA, 2025) clusters. Cluster A comprises 8 nodes and Cluster B consists of 32 nodes, each node is equipped with 8 GPUs.

**MoE Models.** The configurations of the models used are summarized in Table 2. We adopt open-source Qwen-serials (Qwen, 2025) models and customize them for experiments on our setup.

**Baseline Systems.** We compare VRouter against several baselines. First, we evaluate against an internal fork of Megatron-LM (Shoeybi et al., 2019) designed for MoE training, denoted as Megatron-LM+, which integrates multiple parallelism strategies including data and expert parallelism. We also

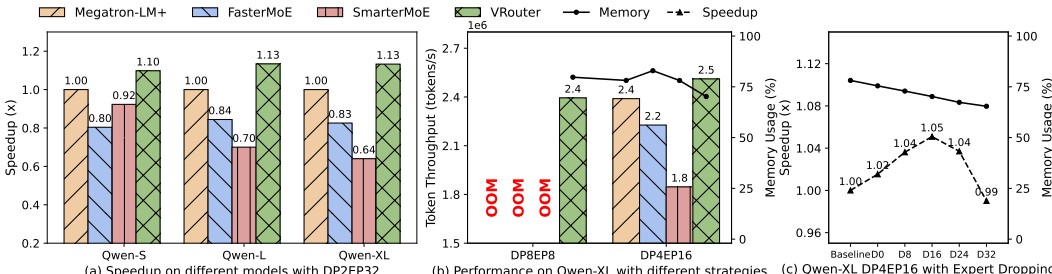

Figure 5: Performance Evaluation on Cluster A. We evaluate VRouter and the baseline systems on Cluster A using 64 GPUs with DP2EP32. "DP2EP32" denotes a data parallelism degree of 2 and an expert parallelism degree of 32. Figure 5(a) illustrates the speedup of VRouter over Megatron-LM+. Figure 5(b) reports the throughput and memory usage of VRouter and the baselines. Figure 5(c) presents both the speedup over Megatron-LM+ and the memory usage of VRouter when dropping different numbers of experts; for example, "D8" denotes the setting $D = 8$.

compare VRouter with FasterMoE and SmartMoE, two state-of-the-art open-source frameworks for load-balancing in MoE training. To ensure a fair and consistent evaluation under modern workloads, we integrate both FasterMoE and SmartMoE into our framework. FlexMoE (Nie et al., 2023), Hecate (Qing et al., 2025) and SwiftMoE (Skiadopoulos et al., 2025) are not publicly available for reproduction, and hence we exclude them from our experiments.

**Evaluation Metrics.** Following prior work (Xue et al., 2025), we use training throughput, measured in tokens per second (tokens/s), as a key metric to evaluate the efficiency of MoE training—the higher the throughput, the faster the model processes training data. We also report memory utilization, defined as the ratio of peak GPU memory consumption to the total available GPU memory capacity, to assess memory efficiency during training.

## 6.2 END-TO-END PERFORMANCE

We first evaluate VRouter and its baselines on three variants of the Qwen families—Qwen-S, Qwen-L, and Qwen-XL—with DP2EP32 parallel configuration (Figure 5(a)). Both FasterMoE and Smart-MoE suffer significant performance degradation, with throughput drops of up to 20% and 36%, respectively, due to the high communication overhead from frequent cross-device expert rearrangement. In contrast, VRouter leverages Inter-EP Routing to achieve fine-grained load balancing at the micro-batch level without requiring any physical movement of experts. This design enables VRouter to deliver consistent performance gains, achieving a 1.10-1.13× speedup over Megatron-LM+ across all model scales.

We then evaluate the effectiveness of VRouter under different sizes of EP group on Qwen-XL (Figure 5(b)). In the DP8EP8 configuration, where each device hosts a large number of experts, the disproportionate traffic to "hot" experts creates severe load imbalance, resulting in excessive activation memory consumption and ultimately causing all baseline systems to fail with out-of-memory (OOM) errors. In contrast, VRouter mitigates this issue through expert dropping and dynamic workload rebalancing via Inter-EP Routing, effectively reducing peak memory usage. As a result, VRouter successfully completes training while utilizing only 79.73% of the total GPU memory capacity. When scaling the EP group size to 16, VRouter further improves throughput by 4.16% compared with DP8EP8, reduces GPU memory consumption to 89% of the baseline's footprint, and achieves a 1.05× speedup over Megatron-LM+, demonstrating strong efficiency and scalability in large-scale MoE training.

To evaluate performance at larger scale, we train Qwen-XXL on Cluster B using 256 GPUs with a parallel configuration of DP=2, TP=2, PP=4 and EP=16 as depicted in Table 3. Here, TP denotes tensor parallelism (Shoeybi et al., 2019), and PP refers to pipeline parallelism (Huang et al., 2019). VRouter achieves a 1.051× speedup over Megatron-LM+, highlighting its effectiveness and compatibility in large-scale MoE training.

## 6.3 ANALYSIS OF EXPERT DROPPING IN VROUTER

We conduct an ablation study to evaluate the impact of expert dropping in VRouter, as shown in Figure 5(c). We vary the hyperparameter $D$ across values 0, 8, 16, 24, and 32, and measure the resulting speedup over Megatron-LM+ along with GPU memory usage.

Table 3: Evaluation of Qwen-XXL on Cluster B with DP=2, TP=2, PP=4 and EP=16. We measure VRouter 's throughput with D=8, i.e. dropping eight experts per device using the expert dropping mechanism.

| System | Config | Throughput($10^6$ tokens/s) | Memory Util. | Speedup |
|---|---|---|---|---|
| Megatron-LM+ | DP2TP2PP4EP16 | 2.219 | 59.574% | 1.000× |
| VRouter | DP2TP2PP4EP16-D8 | **2.333** | **56.160%** | **1.051×** |

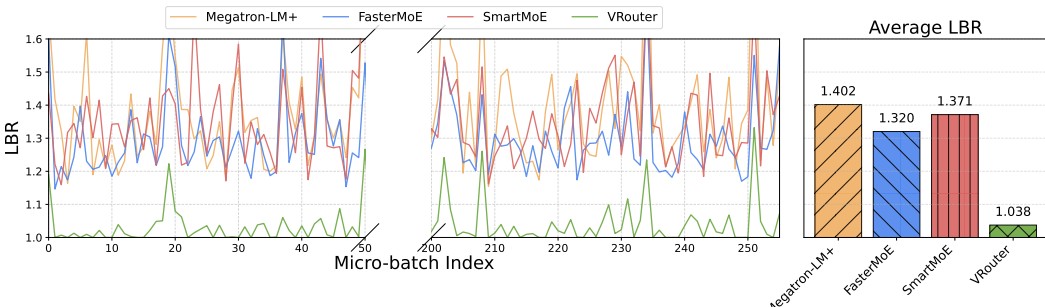

Figure 6: Load balance ratio across micro-batches within a global-batch for different MoE training systems. The data is collected at training iteration 125,020 on Qwen-S in layer 0 with DP2EP32.

As $D$ increases from 0 to 32, memory consumption steadily decreases from 75.47% to 65.42%, demonstrating that expert dropping effectively reduces memory footprint. When $D$ increases from 0 to 16, the speedup improves from 1.02× to 1.05×, indicating that reducing the number of active experts enhances computational efficiency. This gain stems from two factors: (1) each remaining expert processes more tokens on average, improving utilization, and (2) fewer experts reduce the communication overhead associated with synchronizing expert parameters and gradients within the EP group. However, further increasing $D$ beyond 16 leads to diminishing returns. As $D$ reaches 32, the speedup begins to decrease due to the excessive load imbalance introduced by removing too many experts. This imbalance disrupts routing efficiency and degrades both computation and communication performance. Notably, even when no experts are dropped ($D = 0$), VRouter still uses less GPU memory than Megatron-LM+. This reduction is attributed to the inherently better load balancing achieved by Inter-EP Routing, which minimizes activation spikes and promotes more uniform resource utilization across devices.

### 6.4    EXPERIMENT ON LOAD BALANCE RATIO

In Figure 6, we compare the average $LBR$ across several load-balancing systems within a single global batch. In the left plot, the three baselines fluctuate between 1.15 and 2.14. On average, Megatron-LM+ achieves an $LBR$ of 1.4, indicating that the most heavily loaded device processes about 40% more tokens than the average. This imbalance remains serious because SmartMoE rearranges expert placement only at the global-batch level, failing to capture the fine-grained, dynamic load variations that occur during training. As a result, the "hot" device still computes 37.1% more tokens than the average. FasterMoE, which can operate at the micro-batch level, achieves better results than SmartMoE; however, it replicates only a very small subset of experts (on average 3 out of 128) to devices in other EP groups based on a cost model. Consequently, its ability to improve load balance remains limited. In contrast, VRouter reduces the $LBR$ to 1.038 in this experiment—very close to the ideal value—outperforming both SmartMoE and FasterMoE.

### 7    CONCLUSION

In this paper, we present VRouter, a routing system that achieves micro-batch level load balance in MoE model pretraining. The key insight is that load imbalance can be mitigated through Inter-EP Routing by dispatching tokens across multiple EP groups rather than migrating or replicating experts. Building on this idea, VRouter introduces three techniques: (1) an expert shift strategy that modifies expert placements to redistribute workloads across paired EP groups; (2) an expert dropping mechanism that selectively discards experts to reduce memory usage and synchronization overhead while maintaining workload balance; and (3) a load-aware token rerouting algorithm that constructs balanced routing maps via a greedy search based on real-time device load distributions. Experiments demonstrate that VRouter outperforms existing MoE training systems by up to 1.13× in training efficiency.

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
