# OpenReview forum: "VRouter: Micro-batch Level Load Balance via Inter-EP Routing for MoE Training"
_ICLR.cc/2026/Conference — Submitted to ICLR 2026_

### Official Review · Reviewer_1LQs · 2025-10-23

**Soundness:** 2
**Presentation:** 2
**Contribution:** 2
**Rating:** 4
**Confidence:** 4

**Summary:**

This paper analyzes the load balancing problem in MoE models from a micro-batch level. The core design is that alleviating load imbalance through an Inter-EP routing mechanism by distributing tokens across multiple EP components instead of migrating or replicating experts. The authors devise a simple and efficient algorithm for load scheduling. Experiments demonstrate that VRouter achieves 1.13× higher training efficiency than existing MoE training systems.

**Strengths:**

1. The author examines load balancing in the Moe architecture from the perspective of micro-batches, presenting a valuable research topic.

2. The author thoroughly discusses the problems and motivations behind load balancing under mini-batches.

3. This paper employs a method of scheduling experts across expert groups and designs a simple yet effective strategy to address the load balancing problem.

**Weaknesses:**

1. Traditional data parallelism does not introduce dependencies between nodes during forward and backward propagation. However, employing inter-EP communication introduces dependencies between different EP groups. Specifically, the parallel computation efficiency of each layer is constrained by the slowest EP group.

2. The overall organization of this paper could be improved. The author might consider adjusting the placement of some figures (fig.1, fig.2 and fig.3) to help readers better connect them with the surrounding context.

3. The communication latency is a critical factor in EP, particularly during inter-EP scheduling. Therefore, this aspect should be considered when designing expert scheduling algorithms.

4. This work involves both intra-EP and inter-EP communication, so it is necessary to describe the different communication rates in detail in the Testbed section.

5. The work introduces a hyperparameter D that affects overall training efficiency and memory consumption, particularly causing training slowdown as described in Experiment 6.3. However, the authors do not provide a clear algorithm for determining its value.

6. This paper addresses load balancing from the perspective of micro-batches, though this approach appears to have been discussed previously. We hope the authors will compare the differences between this paper and the following works. Demonstrating these distinctions through experiments would be preferable.
---
- Zeng Y, Huang C, Mei Y, et al. EfficientMoE: Optimizing Mixture-of-Experts Model Training With Adaptive Load Balance[J]. IEEE Transactions on Parallel and Distributed Systems, 2025.
- Li J, Sun Z, He X, et al. Locmoe: A low-overhead moe for large language model training[J]. arXiv preprint arXiv:2401.13920, 2024.

**Questions:**

1. What is the complete experimental environment for this testbed?
2. Should communication be incorporated into the algorithm design to better schedule tokens across different expert groups?
3. Is this paper the first to analyze load balancing in the MOE architecture from a micro-batch perspective?

---

> ### Author Response · Authors · 2025-11-28
> **Rebuttal by Authors**
>
> Thanks for your thoughtful and constructive reviews. We provide detailed responses to clarify concerns as follows.
>
> **Question 1: The complete experimental environment for this testbed.**
>
> Our baselines already leverage DeepEP and DualPipe to effectively hide communication overhead. The experiments are conducted on an NVIDIA H800 cluster, with each node hosting 8 H800 GPUs interconnected via 400 GB/s NVLink and equipped with 8 × 400 Gbps InfiniBand links. The largest configuration used in our evaluation scales up to 256 GPUs.
>
> **Question 2 & Weakness 3：The necessity of considering the factor of communication in the algorithm design.**
>
> To rapidly assess the impact of communication overhead, we estimate the upper bound of throughput improvement achievable through communication optimizations. To this end, we construct a Mock-Router on top of the Megatron-LM+ framework. The Mock-Router restricts top-k routing within each EP group and distributes tokens uniformly at random, entirely disregarding training accuracy. Under this intra-EP group routing constraint, it achieves a load-balancing level comparable to that of VRouter. We conduct comparative experiments between Mock-Router and VRouter using the Qwen-XL model under the DP4EP16 configuration. As shown in the following table, the results indicate that such communication optimizations can yield up to a 2.6% throughput improvement. This finding, inspired by your suggestion, provides valuable insights for large-scale MoE pre-training tasks and will be further explored in our future work.
> | System | End-to-end elapsed time (ms)  | Speedup  |
> |------|------|------|
> | VRouter (Inter-EP routing) | 6857  | 1  |
> | Mock-Router (Intra-EP routing)  | 6681  | 1.026  |
>
> **Question 3: Whether this paper is the first one to analyze load balance issue at micro-batch level.**
>
> We are not the first to investigate load distribution at the micro-batch level—prior works such as GShard and EfficientMoE have identified this challenge and sought to mitigate load imbalance by tuning expert capacity. However, these methods often incur a loss in training accuracy, typically as a consequence of token dropping or restrictive routing policies. In contrast, our work is the first to effectively resolve micro-batch-level load imbalance without any degradation in accuracy. A more comprehensive discussion of these related approaches will be provided in the revised version of the paper.
>
> **Weakness 1: The parallel computation efficiency of each layer is constrained by the slowest EP group.**
>
> Our primary goal is to address the critical bottleneck caused by the slowest EP group. To this end, VRouter connects all EP ranks into a logical ring and employs a dynamic routing algorithm that, based on real-time load conditions, reroutes tokens from overloaded devices to underloaded ones. Consequently, the peak load in both EP groups is lowered and aligned more closely with the average, substantially mitigating straggler issues—even if minor imbalances remain—compared to systems without inter-EP routing.
>
> **Weakness 2: Writing improvement.**
>
> Thank you very much for your valuable suggestions on improving our writing. We will incorporate your comments and revise the manuscript accordingly in the final version.
>
> **Weakness 4: Describe the communication rates of both Intra-EP and Inter-EP communication.**
>
> The effective EP communication rate is affected by load balance, so it varies from one micro-batch to another in real workloads. We measured the average algorithm bandwidth over one iteration of QWEN-XL under a DP4EP16 parallelism configuration: the Intra-EP communication bandwidth is 64 GB/s, and the Inter-EP communication bandwidth is 70 GB/s.
>
> **Weakness 5: provide a clear algorithm for determining the num of dropping experts.**
>
> Excessive expert dropping can lead to degraded training throughput due to load imbalance. To mitigate this, we introduce a hyperparameter $D$ that controls the number of experts to be dropped, and we determine its throughput-optimal value prior to training. Specifically, before training, we conduct a grid search over candidate values of $D$. For each candidate, we execute the task for 10 iterations, measure the execution cost, and select the value of $D$ that minimizes this cost as the expert-dropping configuration. The search can be completed in under 30 minutes, which is a practical overhead in our setting, especially considering that full training runs last for several months. Hence, the overhead of searching $D$ is negligible over the entire training cycle.

---

> ### Author Response · Authors · 2025-11-28
> **Rebuttal by Authors**
>
> **Weakness 6: Compare VRouter with EfficientMoE and Locmoe.**
>
> Thanks for pointing out two related works we had not discussed: EfficientMoE and Locmoe. Since their code is not publicly available, we do not include experimental comparisons. Both are intra-EP methods and incur training accuracy loss: EfficientMoE assigns an optimal capacity to each expert and drops excess tokens, while Locmoe adds a locality loss to guide the gate network for more balanced and communication-efficient routing. In contrast, VRouter is an inter-EP routing method that balances load across pairs of EP groups by rerouting tokens between them without degrading model quality. We further enhance inter-EP routing with expert shifting and expert dropping, both performed once before training during expert initialization, so there is no runtime overhead from expert migration. Expert shifting guarantees global connectivity among devices across two EP groups, enabling workload migration from overloaded to underutilized devices. Expert dropping removes redundant expert replicas while keeping at least one copy of each expert, preserving correctness while reducing memory and synchronization cost. To avoid throughput degradation from excessive expert dropping, we introduce a hyperparameter D that controls the number of dropped experts, tuned via grid search. We will add a more detailed discussion of EfficientMoE and Locmoe in the final version.

---

### Official Review · Reviewer_dPj1 · 2025-10-30

**Soundness:** 3
**Presentation:** 3
**Contribution:** 2
**Rating:** 4
**Confidence:** 4

**Summary:**

This paper introduces VRouter, a novel Inter-EP routing system that improves load balance at the micro-batch level in Mixture-of-Experts (MoE) models, without expert migration or replication, by employing expert shifting and dropping techniques to optimize GPU efficiency and reduce communication overhead.

**Strengths:**

1. The experiments are extensive, covering a wide range of situations and scenarios.

**Weaknesses:**

1. One potential drawback is that the paper’s approach can be seen as a variation of MOE folding, which decouples the parallelism formulation of the MOE component. This may limit the novelty of the proposed scheduling method.
2. While the low overhead is a strength, the corresponding end-to-end training improvement is also limited to around 5%.
3. Another consideration is how this approach composes with other scheduling optimizations. The performance improvement from the proposed method may not be additive with those from orthogonal techniques like DeepEP or compute/communication overlap, potentially limiting its practical impact in optimized systems.

**Questions:**

see weakness pls

---

> ### Author Response · Authors · 2025-11-15
> **Seek clarification on a few points raised in the review**
>
> We are sincerely grateful to Reviewer dPj1 for devoting time to review our work and providing invaluable feedback. Prior to submitting our formal rebuttal, we kindly request clarification on several aspects of the review.
>
> 1. Weakness 1 states: “One potential drawback is that the paper’s approach can be seen as a variation of MOE folding, which decouples the parallelism formulation of the MoE component.” We would greatly appreciate it if you could provide a concrete reference of “MoE folding” to help us better understand and thoroughly analyze this point.

---

> > ### Comment · Reviewer_dPj1 · 2025-11-16
> >
> > Thank you for your quick comment. The reference to "MoE folding" can be found in the following paper: [MoE Parallel Folding](https://arxiv.org/abs/2504.14960).

---

> ### Author Response · Authors · 2025-11-28
> **Rebuttal by Authors**
>
> We are grateful for your insightful comments and suggestions, which have helped us identify ways to clarify and improve the paper. We provide detailed responses to clarify concerns as follows.
>
> **Weakness 1: Limited novelty compared with MoE parallel folding.**
>
> The paper MoE Parallel Folding introduces two primary techniques. First, it decouples the parallel configurations of the attention and MoE modules, allowing each to adopt its own optimal parallelism strategies. Second, its flexible token-level dispatcher enables token routing under these independently chosen parallelisms and supports both token-dropping and token-dropless training regimes. However, the dispatcher in MoE Parallel Folding still performs only intra-EP routing and therefore cannot resolve micro-batch–level load imbalance. In contrast, VRouter’s novelty lies in its load-aware inter-EP routing, which enables traffic redistribution across EP groups to rebalance device-level load while preserving the semantics of the underlying parallel scheme. In this respect, VRouter is orthogonal to the optimizations proposed in MoE Parallel Folding.
>
> **Weakness 2: Limited end-to-end improvement.**
>
> Without changing algorithmic results, we evaluated VRouter’s performance under a state-of-the-art framework and high-end hardware. The experiments are conducted in an NVIDIA H800 cluster with 8 H800 GPUs per node equipped with 400 GB/s NVLink as well as 8×400 Gbps InfiniBand. Besides, we have already enabled DeepEP and DualPipe to mitigate communication overhead. In this already highly optimized setting, a 5%–13% improvement is substantial and leads to significant cost savings for large-scale, months-long training on thousands of GPUs. Furthermore, Inter-EP routing offers a complementary perspective that is compatible with frontier EP training frameworks, communication libraries, and gating networks, while preserving numerical accuracy and effectively reducing load imbalance at micro-batch level.
>
> **Weakness 3: The effectiveness of VRouter in optimized systems with DeepEP or compute/communication overlap.**
>
> We have enabled both DeepEP and DualPipe in our system. All reported VRouter performance numbers are therefore measured on an already optimized software–hardware stack, and VRouter still delivers a 5%–13% improvement over the baseline in real industrial production environments. Thank you for pointing this out, we will include additional details about hardware configuration and framework-level optimizations in the revised version.

---

### Official Review · Reviewer_8kQx · 2025-11-01

**Soundness:** 2
**Presentation:** 1
**Contribution:** 2
**Rating:** 2
**Confidence:** 4

**Summary:**

This work propose inter-EP expert placement and routing to optimizing MoE model training by addressing the expert load imbalance at the micro-batch level. The experimental results show that the implemented methods achieve throughput improvements over existing MoE training framework.

**Strengths:**

1. It targets to address the issues commonly observed.
2. The results show that the proposed method yields benefits.

**Weaknesses:**

1. The paper should better situate its work in the context of recent literature. Prior work has mostly focused on load imbalance across experts at the micro-batch level. In contrast, there is a growing, more recent trend of addressing imbalance across devices and at the global-batch level, which the authors should acknowledge.
2. The paper's assumed baseline requires further justification. Existing frameworks for hybrid parallelism already support EP both within and across DP groups. The authors should verify whether their chosen baseline is appropriate, as it appears it could lead to the creation of redundant EP groups.
3. The paper would benefit from a more detailed analysis of the target problem scenario. Specifically, the authors should elaborate on the constraints imposed by throughput and memory requirements, while also considering the impact of heterogeneous communication bandwidths (both inter-node and intra-node) on the proposed solution. It is more valuable than the general discussion on "MOTIVATION, OPPORTUNITIES AND CHALLENGES”.
4. The authors should clarify the novelty of their proposed methods. “Expert shifting" and "expert dropping" seem to be new terms for established expert placement strategies like expert migration and replication. The paper proposes building a termed inter-EP group process, which seems to be only a different framework implementation, compared to existing methods of building an EP group to support a variable number of experts and flexible placement.
5. Reference repeat: "Demons in the detail: On implementing load balancing loss for training specialized mixture-of-expert models".

**Questions:**

Please refer to the issues in the Weakness.

---

> ### Author Response · Authors · 2025-11-15
> **Seek clarification on a few points raised in the review**
>
> We sincerely appreciate Reviewer 8kQx’s insightful feedback. Before submitting our formal rebuttal, however, we would like to seek clarification on a few points raised in the review.
>
> 1. To the best of our knowledge, prior work—such as FasterMoE, SmartMoE, Hecate, and FlexMoE—has primarily addressed load imbalance at the global-batch level. In contrast, VRouter specifically targets load imbalance at the micro-batch level. However, Weakness 1 states that “Prior work has mostly focused on load imbalance across experts at the micro-batch level.” We would greatly appreciate it if you could provide references to works that indeed address load imbalance at micro-batch level.
>
> 2. In our testbed with 64 GPUs, setting DP = 2 and EP = 32 means that the attention layers effectively use full data parallelism across all 64 GPUs, while the MLP layers—which contain the experts—are distributed with data parallelism size of 2 and expert parallelism size of 32. Weakness 2 states that “Existing frameworks for hybrid parallelism already support EP both within and across DP groups” We would greatly appreciate it if you could clarify the relationship between EP and DP groups in more detail—particularly how EP is implemented across DP groups—and provide references to the frameworks that offer such functionality.

---

> > ### Comment · Reviewer_8kQx · 2025-11-27
> >
> > 1. In the initial phase of the EP proposal, the focus is on local group tokens for efficient parallel processing, which conducts expert capacity to limit the load imbalance, without synchronizing to manage the load of global tokens.
> > You can check: Lepikhin, Dmitry, et al. "Gshard: Scaling giant models with conditional computation and automatic sharding." arXiv preprint arXiv:2006.16668 (2020).
> >
> > 2. You can check the implementation in the Megatron repository, with a particular focus on MoE Parallel Folding: https://github.com/NVIDIA/Megatron-LM/tree/main/megatron/core/transformer/moe

---

> ### Author Response · Authors · 2025-11-28
> **Rebuttal by Authors**
>
> We sincerely appreciate your careful reading of our manuscript and insightful comments, which have greatly helped us improve the quality of this work. We provide detailed responses to clarify concerns as follows.
>
> **Weakness 1: The paper should better situate its work in the context of recent literature.**
>
> Thank you for the suggestion. The recently proposed Global-batch Load Balance Loss (see: https://aclanthology.org/2025.acl-long.249/) has been widely adopted and effectively addresses load imbalance at the global-batch level. Although these works addresses load imbalance at the global batch level, micro-batch imbalance still persists. Based on the reviewer’s suggestion, we examined methods such as GShard, which impose an expert capacity and drop excess tokens once the limit is reached, potentially degrading training accuracy. In contrast, VRouter leverages expert shifting to create opportunities for inter-EP load redistribution and, together with Algorithm 1, iteratively drives the system toward balanced token assignment. We will discuss the relationship between VRouter and these token-dropping approaches in the final version.
>
> **Weakness 2:The paper's assumed baseline requires further justification.**
>
> VRouter and all baseline systems in our experiments are implemented on top of Megatron-LM, using commonly adopted combinations of parallelism strategies, inspired by DeepSeek-V3(https://arxiv.org/html/2412.19437v1) and Kimi K2(https://arxiv.org/abs/2507.20534).
>
> **Weakness 3: Provide detailed analysis of the target problem scenario.**
>
> Our target problem setting is to maximize the end-to-end training throughput of MoE models without changing training accuracy, parallelism configurations, or data-related policies (including micro-batch size and global batch size), while ensuring that the runtime memory footprint does not exceed the GPU memory capacity. Although the interconnect topology does have an impact, it does not affect the design of our expert shifting and expert dropping mechanisms. Our core novelty lies precisely in these mechanisms, which create additional opportunities for token redistribution under inter-EP routing. The concrete effect of topology manifests in the routing algorithm in Algorithm 1. At present, this routing algorithm is load-aware but topology-agnostic; incorporating topology awareness is an important direction that we leave to future work.
>
>
> **Weakness 4:  Clarify the novelty of expert shifting and expert dropping.**
>
> Expert shifting and expert dropping are both performed before training, during the initialization of expert parameters. Therefore, no additional overhead from expert migration is introduced during training. Expert shifting guarantees global connectivity among devices, ensuring that there is always a feasible path between any two devices within two EP groups. This connectivity enables workloads to be migrated from overloaded devices to underutilized ones. We can safely drop some expert replicas while still ensuring that each expert has at least one remaining copy. This preserves correct token computation, while the removal of redundant replicas reduces memory usage and lowers the cost of parameter and gradient synchronization among experts. However, if too many experts are dropped, throughput can degrade due to severe load imbalance. To mitigate this, our paper introduces a hyperparameter $D$ that controls the number of experts to be dropped, which we tune via grid search. In contrast, prior approaches based on expert replication or expert migration dynamically change the placement of experts during training. This introduces additional migration overhead, and expert replication further increases memory consumption.
>
> **Weakness 5: Repeated references.**
>
> Thank you very much for pointing out this oversight in our writing. We will carefully check for and remove redundant citations in the revised version of the paper.

---

### Official Review · Reviewer_fE7h · 2025-11-08

**Soundness:** 3
**Presentation:** 3
**Contribution:** 3
**Rating:** 6
**Confidence:** 3

**Summary:**

This paper proposes VRouter that tackles micro-batch-level load imbalance in large-scale MoE training without requiring expert migration or replication.  VRouter introduces three key techniques: (1) Cyclic Expert Shifting, which pairs Expert Parallel (EP) groups and cyclically shifts expert placements in one group to create heterogeneous layouts, enabling tokens to be offloaded across groups;  (2) Expert Dropping, which reduces per-device memory and gradient synchronization overhead by dropping a configurable number of experts while preserving load balance;  and (3) a lightweight load-aware token rerouting algorithm that greedily redistributes tokens from overloaded to underloaded devices based on real-time load information.  Evaluated on Qwen-series MoE models (Qwen-S/L/XL/XXL) against baselines like Megatron-LM+, FasterMoE, and SmartMoE, VRouter achieves 1.05–1.13× higher training throughput, reduces GPU memory usage by up to ~10%, attains a near-ideal Load Balance Ratio (LBR) of 1.038 (vs. 1.32–1.40 for baselines), and avoids out-of-memory errors in large EP configurations, demonstrating superior efficiency, memory optimization, and scalability.

**Strengths:**

* Unlike existing methods that only focus on global-batch level optimization and fail to capture dynamic load variations across micro-batches, VRouter addresses micro-batch level load imbalance.
* VRouter avoids the high communication and memory overhead of moving or copying experts by instead enabling Inter-EP token routing
* VRouter’s load-aware token rerouting algorithm is lightweight, collecting real-time load data and using greedy search for fast routing decisions.
* VRouter works efficiently at scale and integrates seamlessly with existing MoE training pipelines without requiring expert redesign or complex scheduling.

**Weaknesses:**

* Although VRouter uses cyclic expert shifting to create heterogeneous EP group layouts, it still faces the inherent challenge of homogeneous load patterns across EP groups.
* While Expert Dropping reduces memory and communication, dropping too many experts leads to degraded throughput due to increased load imbalance.
* It is unclear how VRouter performs at multi-thousand-GPU scale, where Inter-EP communication across nodes may introduce significant latency or bandwidth bottlenecks.

**Questions:**

Although the rerouting algorithm is described as “lightweight,” the paper does not quantify its runtime overhead (e.g., CPU scheduling time, latency per micro-batch), which could become non-negligible under very small micro-batches or large expert counts.

---

> ### Author Response · Authors · 2025-11-28
> **Rebuttal by Authors**
>
> Thanks for your thorough evaluation and constructive feedback. We provide detailed responses to clarify concerns as follows.
>
> **Questions 1: The latency of lightweight rerouting algorithm especially under very small micro-batch size and large expert counts.**
>
> We implemented a lightweight, GPU-resident routing kernel via CUDA C. To benchmark the overhead under latency-sensitive settings (micro-batch size of 1, 512 experts), we compared the kernel's execution time against a single Transformer layer, which incurred a latency of only 0.085 ms compared to the layer's 40.3 ms. This constitutes a mere 0.21% runtime overhead, which is negligible in practice.
>
> **Weakness 1: The effectivess of expert shift is still limited by the homogeneous load patterns.**
>
> The load distribution is indeed highly irregular, and the imbalance cannot be eliminated in a single iteration of expert shifting. Nevertheless, expert shifting guarantees global connectivity among devices, ensuring that there is always a feasible path between any two nodes. This connectivity creates opportunities to migrate workloads from overloaded devices to underutilized ones. To substantially mitigate load imbalance, however, a load-aware token routing algorithm—such as the one proposed in our paper—is still required.
>
> **Weakness 2: Dropping too many experts leads to degraded throughput.**
>
> Yes. If too many experts are dropped, throughput can degrade due to severe load imbalance. To address this, our paper introduces a hyperparameter $D$ that controls the maximum number of experts that may be dropped, and we tune it via grid search. For each candidate value of $D$, we run the workload for 10 iterations, measure the end-to-end throughput, and select the value that yields the highest throughput as our expert-dropping configuration.
>
> **Weakness 3: The performance of VRouter at multi-thousand-GPU scale.**
>
> VRouter implements inter-EP routing strictly between pairs of EP groups. Even when the system scales to thousands of GPUs, the routing domain of VRouter remains bounded to communication across only two EP groups. In practice, a typical EP group size is 16 (e.g., Kimi K2: https://arxiv.org/pdf/2507.20534v1). Consequently, with an EP size of 16, VRouter’s effective communication scope is restricted to at most 32 devices, regardless of the global GPU count. This configuration has already been empirically validated in our experiments.

---

### Meta-Review · Area_Chair_64zd · 2025-12-31

**Summary:**

The paper proposes VRouter to address micro-batch load imbalance in MoE training via Inter-EP routing, expert shifting, and dropping. While the approach shows some throughput gains (5-13%), the consensus is for rejection. The primary concerns driving this decision are:

Baseline Validity: The premise that standard MoE is restricted to "Intra-EP" routing is contested. Reviewers argue that modern frameworks (e.g., Megatron-LM) already support flexible topologies, making the proposed solution an incremental fix for a self-imposed constraint rather than a fundamental innovation.

System Robustness: Introducing Inter-EP dependencies couples independent EP groups, creating significant straggler risks. The system becomes bottlenecked by the slowest group, introducing fault tolerance issues that outweigh the marginal gains.
Limited Impact: The gains are viewed as marginal relative to the increased system complexity and risks.

**Reviewer Concerns:**

**Addressed Concerns:**

**Runtime Overhead:** The authors successfully clarified that the lightweight rerouting algorithm incurs negligible overhead (0.21%).

**Scaling Scope:** The concern regarding multi-thousand GPU scaling was addressed by clarifying that VRouter's routing domain is bounded to pairs of EP groups (e.g., 32 devices) regardless of global scale.


**Outstanding Concerns:**

**Baseline & Novelty (Critical):** Reviewer 8kQx's concern that the "Intra-EP" baseline is a strawman remains unresolved. The proposed "Expert Shifting" is viewed as a reinvention of existing flexible placement strategies available in current frameworks.

**Straggler Problem:** Reviewer 1LQs's concern about Inter-EP communication introducing dependencies remains a major drawback. The coupled nature of the design means the entire system's speed is dictated by the slowest EP group, reducing robustness compared to independent data-parallel execution.

**Marginal Improvement:** Reviewer dPj1 noted that a 5% improvement is limited , and it is unclear if these gains persist when combined with orthogonal optimizations like DeepEP or communication overlap.

**Reviewer Scores:**

Reviewer 8kQx (Current: 2): Remains 2. Would maintain Strong Reject as they remained unconvinced by the authors' defense regarding the baseline validity.

Reviewer fE7h (Current: 6): 6 to 4. Would likely lower score after seeing the valid concerns raised by Reviewer 8kQx regarding the "strawman" baseline comparisons.

Reviewer dPj1 (Current: 4): Remains 4. Would maintain the score as the rebuttal did not resolve the concern that the method is a minor variation of MoE folding with limited gains.

Reviewer 1LQs (Current: 4): Remains 4. Would maintain the score as the reliance on Inter-EP dependencies creates unresolved system robustness risks.

---

### Decision · Program_Chairs · 2026-01-26

Reject